# The Endo-α(1,3)-Fucoidanase Mef2 Releases Uniquely Branched Oligosaccharides from *Saccharina latissima* Fucoidans

**DOI:** 10.3390/md20050305

**Published:** 2022-04-29

**Authors:** Vy Ha Nguyen Tran, Thuan Thi Nguyen, Sebastian Meier, Jesper Holck, Hang Thi Thuy Cao, Tran Thi Thanh Van, Anne S. Meyer, Maria Dalgaard Mikkelsen

**Affiliations:** 1Section for Protein Chemistry and Enzyme Technology, DTU Bioengineering-Department of Biotechnology and Biomedicine, Technical University of Denmark, 2800 Kongens Lyngby, Denmark; vyha@dtu.dk (V.H.N.T.); thuthi@dtu.dk (T.T.N.); jesho@dtu.dk (J.H.); 2NhaTrang Institute of Technology Research and Application, Vietnam Academy of Science and Technology, 02 Hung Vuong Street, Nhatrang 650000, Vietnam; caohang.nitra@gmail.com (H.T.T.C.); vanvvlnt@yahoo.com.vn (T.T.T.V.); 3Department of Chemistry, Technical University of Denmark, 2800 Kongens Lyngby, Denmark; semei@kemi.dtu.dk

**Keywords:** endo-α-1,3 fucoidanase, Fourier transform infrared spectroscopy (FTIR), fucoidan, sulfated oligosaccharide, *Saccharina latissima*, GH107, glycosyl hydrolase

## Abstract

Fucoidans are complex bioactive sulfated fucosyl-polysaccharides primarily found in brown macroalgae. Endo-fucoidanases catalyze the specific hydrolysis of α-L-fucosyl linkages in fucoidans and can be utilized to tailor-make fucoidan oligosaccharides and elucidate new structural details of fucoidans. In this study, an endo-α(1,3)-fucoidanase encoding gene, *Mef2*, from the marine bacterium *Muricauda eckloniae*, was cloned, and the Mef2 protein was functionally characterized. Based on the primary sequence, Mef2 was suggested to belong to the glycosyl hydrolase family 107 (GH107) in the Carbohydrate Active enZyme database (CAZy). The Mef2 fucoidanase showed maximal activity at pH 8 and 35 °C, although it could tolerate temperatures up to 50 °C. Ca^2+^ was shown to increase the melting temperature from 38 to 44 °C and was furthermore required for optimal activity of Mef2. The substrate specificity of Mef2 was investigated, and Fourier transform infrared spectroscopy (FTIR) was used to determine the enzymatic activity (Units per μM enzyme: U*_f_*/μM) of Mef2 on two structurally different fucoidans, showing an activity of 1.2 × 10^−3^ U*_f_*/μM and 3.6 × 10^−3^ U*_f_*/μM on fucoidans from *Fucus evanescens* and *Saccharina latissima*, respectively. Interestingly, Mef2 was identified as the first described fucoidanase active on fucoidans from *S. latissima*. The fucoidan oligosaccharides released by Mef2 consisted of a backbone of α(1,3)-linked fucosyl residues with unique and novel α(1,4)-linked fucosyl branches, not previously identified in fucoidans from *S. latissima*.

## 1. Introduction

Fucoidans are present in the cell walls of brown macroalgae and constitute the most complex class of marine polysaccharides found to date. Fucoidans are mainly composed of a backbone of sulfated fucosyl residue in addition to normally minor amounts of other sugars such as galactose, xylose, glucose, mannose, rhamnose, and uronic (mostly glucuronic) acids [1]. Fucoidans can be classified by their monosaccharide composition and the backbone linkage type [2], which includes α(1,3)-L-fucose linkages (group 1), repeating α(1,3)- and α(1,4)-L-linked fucose residues (group 2), galactofucans/fucogalactans (group 3), or fucoidans with higher amounts of mannose and/or uronic acid residues (group 4). Finally, there are more complex fucoidans with a composition that can include five or more different sugars (group 5) [2]. Fucoidans may furthermore be randomly acetylated and branched. The structure of fucoidans may differ within the same alga, resulting in a mixture of polyanionic molecules that are difficult to separate and structurally characterize [3,4,5]. 

*S. latissima,* or sugar kelp, is gaining ground as a new commercially cultivated brown macroalgae due to its ability to grow in the Northern hemisphere, notably in the North Atlantic [6]. Fucoidans from *S. latissima* are particularly complex. However, the following four different partial structures have been reported to date: sulfated fucan, galactofucan, fucoglucuronomannan, and fucoglucuronan [3].

Fucoidans show promising applications in disease treatment with biological activities such as antioxidants, anticancer [7], anti-inflammatory [8], anticoagulant [9,10], and immune-modular properties [11]. Newer results have identified fucoidans as a promising drug against age-related macular degeneration (AMD), including an effective inhibition of the vascular endothelial growth factor (VEGF) [12]. In addition, certain fucoidan oligosaccharides can induce bone-regeneration in osteoporotic sheep when coated on implants [13,14]. Together, these functional studies show that fucoidans have many potential drug applications that are largely influenced by fucoidan structure and size. Endo-fucoidanases can be used as tools to design particular and uniform fucoidan oligosaccharide structures, to pave the way for understanding structure–bioactivity relationships, and in turn, to help tailor-make specialized fucoidans for distinct bioactivity functions. 

Fucoidan-degrading organisms are mainly found in marine bacteria such as *Flavobacteriaceae* [15] and *Alteromonadales* [16]. These bacteria produce extracellular fucoidanases (EC 3.2.1.-) with specific substrate and linkage preferences that have been classified based on sequence similarities in the Carbohydrate Active enZyme database (CAZy) [17] families GH107 (EC 3.2.1.212) and GH168 (EC 3.2.1.211). The first GH107 member MfFcnA, an endo-α(1,4)-fucoidanase from *Mariniflexile fucanivorans* SW5, was characterized in 2006 [18]. GH107 contains 28 sequences, all bacterial, of which only six have been characterized, and the linkage specificity has been determined for a few members. In addition, the linkage specificity has been described for a few fucoidanases not yet added to CAZy GH107. The α(1,4)-specific fucoidanases include MfFcnA [18], FFA1, and FFA2 from *Formosa algae KM3553* [19,20,21], Fhf1 [22], and Fhf2 from *Formosa haliotis* [23], and FWf1 and FWf2 from *Wenyingzhuangia fucanilytica* CZ1127 [24]. To date, the only characterized GH107 members with α(1,3) specificity are Fda1 and Fda2 from *Alteromonas* sp. SN-1009 [25]. The characterization of the GH107 fucoidanases has demonstrated varying selectivity for not only the glycosidic linkage but also for sulfation, acetylation, and branching patterns. Because of the wide variety of fucoidan structures in nature, many fucoidanases specific for different structures are thought to exist, though only a few have been discovered to date.

The mesophilic marine bacterium *Muricauda eckloniae* (previously known as *Flagellimonas eckloniae*) from the *Flavobacteriaceae* family was originally isolated from the rhizosphere of the brown seaweed *Ecklonia kurome* [26]. In the present study, we functionally characterized a novel GH107 fucoidanase, Mef2, from *M. eckloniae.* Mef2 was shown to be the first fucoidanase acting on fucoidans from *S. latissima* and furthermore, the first fucoidanase releasing a novel branched fucoidan oligosaccharide from *S. latissima* fucoidans. 

## 2. Results

### 2.1. Sequence Analysis of the Mef2 Fucoidanase from M. eckloniae

Through BLAST analysis using known GH107 fucoidanase sequences, the putative fucoidanase Mef2 was identified (RefSeq: WP_055392200.1). Mef2 was categorized as a discoidin domain-containing protein in NCBI. The predicted Mef2 protein was 1067 amino acids long and was found to contain an 18 amino acid long N-terminal signal peptide according to SignalP [27]. Modular architecture analysis of Mef2 by InterProScan predicted various domain types, some of which overlap (Figure 1a). 

As expected from the discoidin categorization, a coagulation factor F5/8 C-terminal (FA58C) domain (IPR000421 from 464–834) was predicted in Mef2. *Dictyostelium discoideum* (slime mold) has a related domain in the cell adhesion protein discoidin. The FA58C domain in Mef2 is likely the reason why Mef2 was categorized as a discoidin domain-containing protein when deposited in NCBI.

A Secretion system C-terminal sorting domain (IPR026444 from 997–1067) or Type IX Secretion System (T9SS) domain was predicted in the C-terminus of Mef2. T9SS domains are normally associated with protein sorting to the outer membrane [28]. These domains have not been associated with catalytic activity in fucoidanases and can readily be removed for heterologous expression [18,19,20,22,23,29]. 

An invasin/intimin cell-adhesion fragment (IPR008964 from 385-680) and a bacterial Ig-like group 2 domain (IPR003343, from 387–480) were also predicted in Mef2. The superfamily comprising the Ig-like domain type, including invasin/intimin cell-adhesion domains, is common in GH107 fucoidanases [18,19,20,22,23]. These Ig-like domains are thought to be calcium-binding, which was verified in the crystal structure of the fucoidanase MfFcnA [30]. Ig-like domains have been hypothesized to play a role in substrate recognition and/or enzyme activity of fucoidanases, although this needs further investigations [18,30]. However, these domains might not be crucial for function in in vitro experiments, since at least one fucoidanase, Fhf2, where the domains had been removed, retained activity [23]. 

Several domains likely involved in carbohydrate binding were furthermore predicted in Mef2, a fucolectin tachylectin-4 pentraxin-1 (FTP1) domain (IPR006585 from 475–845), a galactose-binding-like superfamily domain (IPR000421 from 474–963) and a carbohydrate-binding module CBM6 domain (IPR005084 from 842–963). None of these domain types have previously been identified in fucoidanases, and although they might be involved in carbohydrate binding in fucoidanases, their function remains to be investigated.

The previously identified fucoidanase catalytic D1 domain, comprising a (β/α)_8_-barrel fold [30], has not yet been added to any domain prediction programs but was identified in Mef2 through sequence alignment with MfFcnA and P5AFcnA, from which D1 was determined through crystal structure analysis [30]. Sequence alignments of the catalytic D1 domain of Mef2 and the characterized GH107 fucoidanases revealed identity values ranging from 15–52% (Appendix A), with the highest identity to the P5AFcnA (39.2%) and the α(1,3)-linkage specific Fda1 (21%) and Fda2 (22.5%) fucoidanases [25]. A phylogenetic tree (Figure 1b) constructed based on D1 multiple alignments (Appendix A) showed that Mef2 clusters closely with the fucoidanases with which it shares a high identity, while α(1,4)-linkage specific fucoidanases cluster more distantly, giving indications of a likely α(1,3)-linkage specificity of Mef2.

The conserved GH107 fucoidanase motifs ‘RxxxxxDxxxxD’ and ‘DxxxGH’, including the conserved catalytic site amino acids aspartate and histidine, were both identified within the D1 domain of Mef2 (Table 1); however, in the Mef2 sequence, the conserved ‘RxxxxxDxxxxD’ motif was replaced by ‘VxxxxxDxxxxD’. The catalytic aspartate and histidine residues that are predicted to function as the catalytic nucleophile and acid/base and that are likely responsible for the cleavage of the glycosidic linkages by GH107 fucoidanases [30] were identified as Asp182 and His260 in Mef2 (Table 1 and Appendix A).

The alignment also revealed conserved amino acids in Mef2 (Tyr127, Asn207 and Trp307) (Table 2 and Appendix A), previously predicted to be involved in coordinating the −1 subsite important for fucoidan hydrolysis [30]. However, another asparagine (Asn149 in MfFcnA) showing conservation in many GH107 fucoidanases was a serine (Ser129) in Mef2. Two other characterized fucoidanases, the α(1,3) linkage specific fucoidanases Fda1 and Fda2, also show differences to the canonical Ser129, having an alanine in this site. 

### 2.2. Functional Characterization of the Recombinant Mef2 Fucoidanase

The recombinant Mef2 encoding gene was constructed without the predicted signal peptide and T9SS domain but with a C-terminal 10×his tag, giving a predicted molecular weight of 105 kDa (GenBank: ON099398). Mef2 was co-expressed in *E. coli* BL21 (DE3) with the pGro7 chaperone. Expression of Mef2 at 20 °C overnight resulted in a partially degraded protein (Appendix A), which was however resolved by an expression for 4 h at 37 °C (Appendix A), resulting in one protein band at the expected size of approximately 105 kDa. This might indicate that fast expression of Mef2 at 37 °C in the presence of a chaperone results in better folding and less degradation than slower expression over a longer time at 20 °C. In previous expressions of several fucoidanases with extended C-terminal domains, expression was optimized by C-terminal deletions [22,23,29], which were, however, not necessary for successful Mef2 expression.

The substrate specificity of Mef2 was investigated on fucoidans with different structures isolated from 11 different species of brown seaweeds (Figure 2). Mef2 efficiently catalyzed the hydrolysis of fucoidans isolated from the brown alga *F. evanescens* that consist of alternating α(1,3)- and α(1,4)-linked fucose residues [1], releasing fucoidan oligosaccharides with varying degrees of polymerization (DP), including low molecular weight products. Furthermore, Mef2 also catalyzed hydrolysis of the galactofucans isolated from *Sargassum mcclurei*, *Sargassum polycystum*, *Saccharina cichorioides,* and *S. latissima*, the latter three of which are known to contain α(1,3)- linked α-L-fucose residues [3,31]. Interestingly, Mef2 was the first enzyme found able to efficiently catalyze the hydrolysis of fucoidans from *S. latissima*. Together, results indicate that Mef2 catalyzes hydrolysis of α(1,3) glycosidic bonds in fucoidans, consistent with the phylogenetic clustering. 

The hydrolysis of fucoidans from *F. evanescens* and *S. latissima* by Mef2 was monitored from 0 to 48 h by Carbohydrate-Polyacrylamide Gel Electrophoresis (C-PAGE) and High Performance Size Exclusion Chromatography (HP-SEC) (Figure 3). The small fucoidan oligosaccharides were visible in the C-PAGE after 5 and 30 min reactions on fucoidans from *F. evanescens* and *S. latissima*, respectively, while complete degradation was not achieved before 48 h on *F. evanescens* and 5 h on fucoidans from *S. latissima*. 

The HP-SEC analysis showed that both fucoidan substrates decreased substantially in size already after 5 min of reaction with Mef2. The resulting molecular weight distribution of fucoidans from *F. evanescens* changed from a single broad peak of approximately 350 kDa (100 to >800 kDa) to a polydisperse peak of approximately 5–200 kDa (Figure 3b). Mef2 HP-SEC analysis on *S. latissima* fucoidans revealed a reduction of the most dominant molecular weight from approximately 300 kDa (70 to >800 kDa) to two fucoidan populations, one with the most dominant molecular weight at 200 kDa (70–500 kDa) and a low molecular weight population with a most dominant molecular weight of around 1 kDa (0.6–3 kDa) (Figure 3d). These results showed that the Mef2 catalyzed hydrolysis of fucoidans was initiated by the hydrolysis of larger polymers, as visible by HP-SEC analysis, followed by further hydrolysis into smaller oligosaccharides, as visible by C-PAGE analysis.

The lowest band in C-PAGE resulting from FFA2 catalyzed hydrolysis of fucoidans from *F. evanescens* corresponds to a tetra-saccharide of (1,4)- and (1,3)-linked α-L-fucosyls with each fucosyl residue sulfated at C2 (DP4) [20]. However, the *S. latissima* and *F. evanescens* fucoidan-derived oligosaccharides migrating furthest in the gel after Mef2 catalyzed fucoidan hydrolysis did not co-migrate with the tetra-saccharide of the standard but migrated slightly slower in the gel, suggesting slightly larger and/or less charged oligosaccharides. 

The influence of pH on Mef2 activity using buffers with overlapping pH values (UB4 buffer pH 2–8 and borate buffer pH 8–11) showed that Mef2 was active at pH values from 6 to 8 in the UB4 buffer but not in the borate buffer, while the optimal pH was found at 7–8 (Figure 4a). Mef2 was found active at temperatures ranging from 15–50 °C, while the optimal temperature was between 30 and 37 °C (Figure 4b). Consistent with the C-PAGE results, the melting temperature (Tm) of Mef2 was determined to be 38 °C when Ca^2+^ had been removed by EDTA (Appendix A). For further experiments, 35 °C was used.

Most fucoidanases are divalent cation-dependent enzymes [19,20,22,23,24], which has been supported by fucoidanase crystal structures, where Ca^2+^ was found in the catalytic D1 domain [30]. The metal ion-dependency was therefore investigated for Mef2. The Mef2 enzyme was stripped of divalent cations by EDTA, resulting in a complete loss of function (Figure 4c). Different divalent cations at 10 mM concentration showed varying effects on the Mef2 enzyme, where Mg^2+^, Cu^2+^, Fe^2+^, Zn^2+^, Co^2+^, and Ni^2+^ did not reactivate the Mef2 fucoidanase, while Mn^2+^ resulted in a slight re-activation, and Ca^2+^ showed the best re-activation of Mef2. The effect of Ca^2+^ on the stability of the protein (Tm) showed an increase in Tm of 6 °C in the presence of Ca^2+^, resulting in a Tm of 44 °C compared to 38 °C without Ca^2+^ (Appendix A). 

The influence of NaCl on Mef2 activity showed that Mef2 was not affected by NaCl as the activity did not differ significantly at concentrations from 25 to 400 mM (Figure 4d). Together, the optimal conditions for Mef2 were 0.9% fucoidan, 20 mM Tris-HCl pH 8, 10 mM CaCl_2_, 100 mM NaCl, and 35 °C.

### 2.3. Determination of the Mef2 Fucoidanase Unit by Fourier Transform Infrared Spectroscopy (FTIR)

A Fourier transform infrared spectroscopy was previously used to determine the activity of four different α(1,4) specific GH107 endo-fucoidanases (MfFcnA, Fhf1, Fhf2, and FFA2) on fucoidans from *F. evanescens* [23,32] and three (MfFcnA, Fhf1, and FFA2) as well on fucoidans from *F. vesiculosus* [32]. The enzyme dose-related changes in the FTIR spectrum were followed using the multivariant platform PARAFAC, where the enzyme dose increase is linearly correlated to the PARAFAC score. In this manner, a fucoidanase unit was established, where one enzymatic unit was defined as the amount of enzyme able to increase the PARAFAC value score by 0.01 [32]. A linear equation results from each calibration curve as follows: PARAFAC score = a × concentration of enzyme + b [32].

To determine the fucoidanase unit of Mef2, *F. evanescens* fucoidan degradation was monitored by FTIR (Figure 5). The Mef2 concentration-dependent enzymatic hydrolysis of *F. evanescens* fucoidans (FeF4, Appendix A) showed an increase in absorption for wavenumbers 1150–1200 cm^−1^ (score increasing from 0 to 2) and 1300–1350 cm^−1^ (from 0 to 1), which indicated changes in vibrations of C–O–C stretching of the glycosidic bonds and unidentified bonds, respectively. Furthermore, a decrease in wavenumbers 1200–1250 cm^−1^ (from 0 to −0.5) and 1400–1500 cm^−1^ (from 0 to −3) was observed, indicating changes in vibrations of S=O of the sulfate group and O–C–O of carboxylate group bonds, respectively. Changes in these wavenumbers were also previously observed for the fucoidanases on fucoidans from *F. evanescens* [23,32]. PARAFAC analysis on Mef2 FTIR data resulted in a linear curve with the equation 0.01 = −0.001 × conc Mef2 + 0.0019 and an *R*^2^ value of 0.97 (Appendix A), which led to the specific activity (U*_f_*/μM) of 1.2 × 10^−3^ U*_f_*/μM.



(1)
Mef2: 0.01=−0.001 × concMef2+0.0019⇒concMef2=8.10 µM



Hence, the specific activity of Mef2 (F. evanescens) = 1.2 × 10−3 U*_f_*/μM.

Compared to the units of the previously characterized α(1,4)-specific fucoidanases Fhf1 (1.2 × 10^−3^ U*_f_*/μM), MfFcnA (2 × 10^−3^ U*_f_*/μM), Fhf2 (2.4 × 10^−4^ U*_f_*/μM) and FFA2 (4 × 10^−3^ U*_f_*/μM) on fucoidans from *F. evanescens*, the Mef2 enzyme presumably works slower than FFA2 and MfFcnA, to a comparable level as Fhf1, while faster than Fhf2 [23,32]. 

Since the Mef2 was active on fucoidans from *S. latissima* and this substrate has not previously been subjected to FTIR analysis using fucoidanases, FTIR kinetics was also performed with *S. latissima* fucoidans (SlF4) (Appendix A, Figure 6). The spectral evolution profile resulted in changes in the same wavenumbers as observed for *F. evanescens* fucoidans, but the changes were larger for the *S. latissima* fucoidans. An increase in absorption in wavenumbers 1150–1200 (from 0 to 5) and 1300–1350 (from 0 to 2.5), and decreases in wavenumbers 1200–1250 (from 0 to −6) and 1400–1500 (from 0 to −9), indicated that the overall change in the substrate-product solution was larger for *S. latissima* than for *F. evanescens* fucoidans. PARAFAC analysis on the Mef2 FTIR data resulted in a linear curve with the equation 0.01 = 0.0129 × conc Mef2–0.0257 and an R^2^ value of 0.98 (Appendix A), which led to the specific activity (U*_f_*/μM) of 3.6 × 10^−3^ U*_f_*/μM.



(2)
Mef2 (S.latissima):0.01=0.0129 × concMef2−0.0257⇒concMef2=2.80 µM



Hence, the specific activity of Mef2 (*S. latissima*) was 3.6 × 10^−3^ U*_f_*/μM.

### 2.4. Determination of the Mef2 Linkage Specificity and Structural Elucidation of the S. latissima Fucoidan Products by NMR

Due to the large structural complexity of the fucoidans from *S. latissima*, the fucoidans were deacetylated and verified by NMR analysis (Appendix A) before Mef2 catalyzed hydrolysis. The Mef2 reaction products were separated into the following two fractions: low molecular weight fucoidan products (LMP) and medium molecular weight products (MMP) by ethanol precipitation (Appendix A). The yields of MMP and LMP were 86% and 14%, respectively. While MMP was composed of different monosaccharides, with fucose and galactose in almost comparable amounts (Appendix A), the only monosaccharide detectable in the LMP was fucose. 

The fucoidan digestion using the Mef2 endo-fucoidanase was completed after the first reaction since both the MMP and LMP were not further degraded by the second step of Mef2 hydrolysis (Appendix A). The Mef2 HP-SEC analysis on the *S. latissima* fucoidans revealed that the LMP had an average molecular weight of about 4 kDa, and the MMP had an average molecular weight of about 200 kDa (Appendix A).

For structure investigations, MMP was separated by ion-exchange chromatography (IEX) and pooled according to total carbohydrate content and C-PAGE analysis, resulting in six fractions (MF1-6) (Figure 7a,b and Appendix A). The MF fractions were heterogeneous and contained a mixture of fucoidan oligosaccharides of different sizes (Figure 7d). MF1-6 were additionally analyzed by C-PAGE and did not represent homogenous samples but rather fractions with more than one band, as well as high molecular weight fucoidans not migrating in the gel, indicating that the fractions contained different fucoidan poly- and oligosaccharides (Figure 7c). In particular, the MF fractions contained fucoidans of high molecular weight (around 200 kDa), visual as a band in the top of the C-PAGE gel, too large to migrate, and low molecular weight fucoidans (around 4–10 kDa), migrating as bands in the C-PAGE gel. For MF4 and 5, no low molecular weight fucoidans were observed, but in contrast to MF6 (and the high molecular weight fucoidans in MF1), where the high molecular weight fucoidans could not migrate into the C-PAGE gel, MF4 and 5 contained high molecular weight fucoidans able to migrate slightly into the gel, resulting in a smear at the top of the C-PAGE gel. The slight migration of MF4 and MF5 fucoidans could be discerned in the HP-SEC, where a slight shift to the right (lower molecular weight) could be observed for MF4 and 5 compared to MF6. The monosaccharide composition of the MF fractions was determined (Appendix A). The ratio of fucose:galactose varied but correlated with the average molecular weight of each fraction. The galactose content was higher in the larger-sized fucoidan fractions. 

The structure of fractions MF2 to MF6 was further investigated by NMR spectroscopy. The NMR spectra resembled a mixture of 1-3 linked α-fucosyl units (the small fraction MF2) and β-1,4 as well as β-1,6 linked galactosyl units predominating in the MMP with NMR spectral characteristics corresponding to structures previously described [3]. 

In the higher molecular weight fractions (MF3-MF6), three separate spin systems were observed that resembled spin systems present in the substrate fraction of *S. latissima* (Figure 7d). Two of those spin systems could be assigned to galactose based on the distinct coupling pattern yielding efficient magnetization transfer between H1 and H4, but not between H4 and H5 (see Appendix A). The third spin system was identified as fucose. Chemical shifts and sequential assignments are collected for the deacetylated medium molecular weight samples in Table 3. Linkage patterns were established based on HMBC and NOE correlations, i.e., proximity between pairs of nuclear spins in chemical structure and space across the glycosidic bonds. The structure is consistent with a previously reported structural motif detected in the reinvestigation of desulfated *S. latissima* fractions [3] and with the presence of galactose and fucose in the polysaccharide as determined by monosaccharide analysis. The NMR on the deacetylated sample showed the sulfation pattern displayed in Figure 7d. The NMR spectroscopy on the native, non-deacetylated high molecular weight fraction SlF3 showed a strong and characteristic deshielding of ^1^H3 in unit I (to chemical shifts of 5.260 ppm/71.70 ppm for the C3H3 group) and an equivalent deshielding of ^1^H2 in unit J (to 5.218 ppm/70.68 ppm for the C2H2 group; not shown). Hence, the preparation was predominantly acetylated at O3 of unit I and O2 of unit J.

The LMP were further separated by IEX to obtain pure oligosaccharides. In total, four fractions (OF1-4) were obtained, as determined by total carbohydrate analysis via the phenol-sulfuric acid method [33] (Appendix A) and C-PAGE (Figure 8a). OF1 showed the highest NMR spectral quality (Table 4) and provided sufficient material for a full NMR structure determination. OF2-4 proved to provide some signals resembling OF1 and thus indicating that these oligosaccharides share the same core as the OF1 oligosaccharide. The signals within each spin system of OF1 were assigned primarily based upon ^1^H-^1^H COSY, ^1^H-^1^H TOCSY, ^1^H-^13^C HMBC, ^1^H-^13^C H2BC, and ^1^H-^13^C HSQC correlations. As fucose is a 6-deoxygalactose, it shares the coupling pattern of galactose between hydrogens one and five, with a small coupling of approximately 1 Hz between the hydrogen atoms four and five. Spin systems were thus primarily assigned with TOCSY and heteronuclear NMR to identify CH groups for atoms 1-4 in each fucose residue. C4 could then be correlated to the CH groups five and six through HMBC correlations from the methyl group. A total of ten spin systems were identified and designated A-H, as shown in Table 4. Reducing end signals constituted two of the spin systems, while the adjacent residue likewise yielded spin systems that were distinct for the reducing end anomer on a high-field (800 MHz) NMR instrument. Overall, the oligosaccharide of fraction OF1 was thus a reducing oligosaccharide constituted of eight carbohydrate residues, with the structure α-L-Fucp-(4OSO_3_^−^)-(1,3)-α-L-Fucp-(2OSO_3_^−^)-(1,3)-α-L-Fucp-(1,3)-α-L-Fucp-(4OSO_3_^−^)-(1,3)-α-L-Fucp-(2OSO_3_^−^)-(1,3)-α-L-Fucp-(OSO_3_^−^) with branches α-L-Fucp-(4OSO_3_^−^)-(1→ connected to α-L-Fucp-(2OSO_3_^−^) at C4 of two of the units (second unit from the reducing end and the non-reducing end, respectively).

A sequential assignment of the spin systems was achieved primarily using ^1^H-^13^C-HMBC and ^1^H-^1^H NOESY spectra to detect ^3^*J*_CH_ correlations and NOES across the glycosidic bonds, yielding the sequential assignment shown in Table 4. The structural assignment is consistent with chemical shift data insofar as glycosydically linked positions show a deshielding of their ^13^C chemical shifts but less so with their ^1^H chemical shifts. Six of the eight residues additionally showed a characteristic deshielding of ^1^H positions (to approximately 4.5 ppm) and the corresponding ^13^C signals at these positions, strongly indicative of the attachment of sulfate groups at these secondary alcohol sites. In contrast, acetyl groups were not present, as witnessed by the absence of acetyl ^1^H NMR signals near 2 ppm and of attachment sites at secondary alcohol CH groups with strongly deshielded ^1^H. 

Surprisingly, the oligosaccharide structure showed the branching of the expected α(1,3)-linked backbone at the C4 position, in contrast to previous findings of only C2 branchings in fragments of *S. latissima* fucoidans [3]. Figure 8 exemplifies the identification of the α(1,4) linked branch from the ^1^H-^13^C HSQC and ^1^H-^13^C HMBC spectrum. Beyond correlating to C5 atoms with characteristic chemical shifts near 67 ppm, the methyl protons at C6 also correlate to C4 in the ^1^H-^13^C HMBC spectrums. Two characteristic C4 positions can be identified with deshielded ^13^C nuclei (80.85–81.46 ppm) but non-deshielded ^1^H, indicative of glycosidically linked C4 positions. These C4 positions accordingly show ^3^*J*_CH_ correlations in the ^1^H-^13^C HMBC spectrum to anomeric CH groups of other spin systems (highlighted for one of the residues with branching at C4 in Figure 8c,d). Both H1-C4′ and C1-H4′ correlations across the glycosidic bond are detected for the two C4-branched residues. The C4 branching is further corroborated by the observation of ^1^H-^1^H NOEs between the corresponding H1′ and H4, where the apostrophe designates the terminal residue, as shown in the NOESY spectrum of Figure 8c. Specifically, the 1,4 branchpoint highlighted in Figure 8c is the linkage between residues B and G.

The ^1^H-^13^C spectra from another fraction (OF2) resembled the spectra for OF1 with some differences; however, a new reducing end was formed in this structure, with signals for the OF1 glycosidically linked residue E vanishing, the reducing end signal shifting due to the presence of a new reducing end, and signals for residues D and H exhibiting two sets of signals, indicative of their vicinity to the new OF2 α- and β-reducing end residue E (Appendix A). A possible structure of the OF2 fraction was suggested as a branched sulfated hepta-saccharide (Appendix A), but further experiments are necessary for a full structure determination.

Together, the results verify that Mef2 hydrolyzes fucoidans by endo-α(1,3)-specificity, supported by NMR analysis as well as substrate specificity and phylogenetic clustering. Mef2 is the first characterized fucoidanase shown to be able to hydrolyze the very complex fucoidans from *S. latissima*, thus releasing fucoidan oligosaccharides with a new branch linkage while the galactose containing oligo- and poly-saccharides were retained in the MMP. 

Additionally, the low molecular weight products released by Mef2 from fucoidans from *F. evanescens* were subjected to NMR spectroscopy. The results indicated a backbone of α(1-3)- and α(1-4)-linked fucosyl units at comparable amounts, as follows: the chemical shifts in these fragments were consistent with the linear polymer fraction that was previously isolated as a minor component from fucoidans from *F. evanescens* with enzyme preparations from a marine mollusk [34]. The fucose residues were partly acetylated and devoid of 2,4 disulfations (Appendix A). 

## 3. Discussion

The Mef2 endo-fucoidanase was found to have optimal activity in the neutral or slightly alkaline pH range and an optimal temperature of about 35 °C. Mef2 had optimal activity at salt concentrations of 100–400 mM and was Ca^2+^ dependent. Previous findings showed that the endo-α(1,3) fucoidanases Fda1 and Fda2 have a slightly lower optimal temperature of 30 and 32 °C, respectively [35], whereas most characterized endo-α(1,4) fucoidanases have an optimal temperature of 35–37 °C [18,20,22,23]. Interestingly, Mef2 even showed activity at 50 °C, indicating that Mef2 is more heat stable than Fda1 and 2. Interestingly, the melting temperature of Mef2 was largely affected by the presence of Ca^2+^, resulting in an increase of Tm from 38 °C to 44 °C in the presence of Ca^2+^, consistent with the general Ca^2+^ dependency of known GH107 fucoidanases and the hypothesized function in stabilization of the enzymes [30]. 

The activity of Mef2 was analyzed by FTIR on fucoidans from *F. evanescens* and showed that the spectral changes upon increasing concentrations of enzyme resembled the results obtained from the Fhf1 fucoidanase, giving a specific activity of 1.2 × 10^−3^ U*_f_*/μM and a lower activity than the endo-fucoidanases MfFcnA and FFA2, while a higher activity than Fhf2 [23,32]. In addition, the FTIR assay was evaluated on fucoidans from *S. latissima* for the first time and indicated that Mef2 activity results in larger spectral changes on fucoidans from *S. latissima* than on fucoidans from *F. evanescens*. These spectral changes could be related to changes in vibrations in both the substrate and products. Due to the complex structure of *S. latissima* fucoidans, the large spectral changes might be related to changes in the substrate/product rather than being the direct result of glycosidic cleavage. 

Mef2 interestingly showed activity on both branched and unbranched fucoidans from *S. latissima* and *F. evanescens*, respectively, and was highly selective for fucose, not allowing galactose in the oligosaccharide products, as supported by NMR. NMR analysis showed that MMP were a mixture of galactofucans, resembling previously reported structural motifs from *S. latissima* fucoidans [3]. Together, these findings indicate that Mef2 is selective for fucosyl residues in the backbone and only for α(1,3) linkages, but that the active-site region, likely at the +2 and −2 subsite, would allow for fucosyl branchings. The amino acids in these subsites have not yet been identified in fucoidanases since no crystal structures have been published with substrate bound in the active site. Only the active site and the −1 subsite have been proposed [30]. In the active site, a conserved amino acid has been changed from an arginine to a valine in Mef2, while in the −1 subsite, a conserved asparagine that shows conservation in many other GH107 fucoidanases, was an alanine in the α(1,3) linkage-specific Fda1 and 2 and a serine in Mef2. Whether or not these amino acid changes contribute to the unique specificity of Mef2 requires further investigation.

Further selectivity of Mef2 is supported by the substrate specificity since no activity was detected on other substrates with α(1,3)-linkages, including *F. vesiculosus* or *U. pinnatifida*, indicating that sulfate position and degree of sulfation influence Mef2 activity. In *F. evanescens* C2 and C4, sulfations predominate [36], while fucoidans from *F. vesiculosus* have been found with many different sulfation patterns, including sulfates on C2, C2/C3, C2/C4 or C4 [37,38]. The fucoidans from *U. pinnatifida* are, moreover, assumed to be rich in 2,4-disulfate substitutions [39], while *S. latissima* fucoidans are sulfated at C2 and/or at C4 and C3 [3]. These observations indicate that Mef2 prefers C2 and C4 mono-sulfations and few or no disulfations. Indeed, NMR analysis showed that all the fucosyl residues close to the cleavage site in the oligosaccharides deriving from Mef2 cleavage of *S. latissima* fucoidans are C2 or C4 mono-sulfated.

Some other branched fucoidan oligosaccharides that can be released by fucoidanase activity have been described previously, including the *F. evanescens* fucoidan product of an endo-α(1,4)-fucoidanase from the marine mollusk *Lambis sp*. with the backbone α-L-Fucp-(2OSO_3_^−^)-(1,3)-α-L-Fucp-(2OSO_3_^−^)-(1,4)-α-L-Fucp-(2OSO_3_^−^)-(1,3)-α-L-Fucp-(2OSO_3_^−^) with an α-L-Fucp-(1,4)-branch [40]. In another study, the treatment of fucoidans from *S. horneri* by the FFA1 fucoidanase released branched oligosaccharides with the backbone structure α-L-Fucp-(1,3)-α-L-Fucp-(1,4)-α-L-Fucp-(1,3)-α-L-Fucp and the side chain of α-L-Fucp-(1,2)-α-L-Fucp-1→ at C4 of one main fucosyl residue [19]. Both of these fucoidan oligosaccharides have α(1,3)- and α(1,4)-linked backbones, whereas the Mef2 released oligosaccharides from *S. latissima* have exclusively α(1,3)-linked backbones.

The current study is the first report of fucoidanase activity on fucoidans from *S. latissima* and reports new structural features for fucoidans from *S. latissima*. The LMP were sulfated oligosaccharides of α(1,3)-linked fucose residues and branched at the new C4 position by single mono-sulfated fucose residues. In contrast, the sulfated fucan portion of fucoidans from *S. latissima* was previously only reported to be branched at C2 by single fucose residues [3]. The differences in branching linkage might be related to the cultivated (in this study) versus wild-harvested (in previous studies) *S. latissima* seaweed, or simply the different geographic areas from which they grew, since fine-structure differences are common in fucoidans from different locations [2]. Thus, the use of enzyme hydrolysis provided evidence of the presence of a new branch point as well as branch structure in fucoidans from *S. latissima*. The obtained data indicate that Mef2 can selectively hydrolyze the α-(1,3)-glycosidic bonds in the sulfated fucan part of fucoidans from *S. latissima*.

Only a few of the fucoidanases discovered to date share the exact specificity with regards to glycosidic linkages, branching, and sulfate pattern, while most fucoidanases are unique. This might be expected due to the large structural complexity of fucoidan molecules within and between species of brown seaweed. For a complete degradation of all the different linkages, with different sulfation patterns and branching in fucoidans, a whole battery of hydrolytic enzymes is hence likely necessary, and only a few of these enzymes have been characterized to date.

## 4. Materials and Methods

### 4.1. Fucoidan Substrates

Fucoidans from the brown alga *F. vesiculosus* was purchased from Sigma-Aldrich (Steinheim, Germany). Fucoidans from *S. mcclurei*, *T. ornata*, *S. polycystum*, *H. cuneiformis*, *S. oligocystum*, and *S. serratum* were extracted by a chemical method according to Bilan et al. (2002). Briefly, seaweeds were treated with 0.1 N HCl for 3 h at 70–85 °C to obtain water-soluble polysaccharides. The soluble extract was then treated with 2% CaCl_2_ to remove alginates. The fucoidans were then precipitated from the supernatant by 1% hexadecyltrimethylammonium bromide (Cetavlon, Sigma-Aldrich, Steinheim, Germany). The fucoidan-cetavlon precipitate was isolated by centrifugation and washed with water stirred with 20% ethanolic NaI solution for 2–3 days at room temperature, washed with ethanol, and dissolved in water. The solution was dialyzed, concentrated, and lyophilized [36] and then fractionated by IEX as described previously [33]. For all fucoidan extracts, the polysaccharide fraction eluting the latest, fucoidan fraction 3 (F3), considered the purest fucoidan fraction [33], was used for enzyme experiments. The chemical composition of the individual fucoidans used in the experiments, analyzed as described in [29] with sulfate content determined according to [30], see Section 2.2, is presented in Appendix A.

Fucoidans from *F. evanescens* were extracted using an enzyme-assisted method and further fractionated according to the method previously described [33], with an extra later eluted fraction FeF4 (Appendix A), used for FTIR analysis. In addition, fucoidans from cultivated *S. latissima* seaweeds from Ocean Rainforest were extracted by the same enzyme-assisted method with slight modifications. In short, seaweed was treated by a mixture of the commercial cellulase blend Cellic^®^CTec2 (5% *v*/*w*) and the alginate lyase SALy (0.5% *w*/*w*) (the latter prepared by heterologous expression in *E. coli* as described previously in [41]) in 40 mM Tris-HCl buffer pH 7 at 40 °C for 24 h. The reaction was stopped by incubating at 90 °C for 10 min. The alginate was precipitated by 2% CaCl_2_ and removed by centrifugation. The fucoidans were precipitated by the addition of 96% ethanol (EtOH) in a fucoidan:EtOH (*v*/*v*) ratio of 1:3 and lyophilized. The fucoidan extracts were further separated by DEAE-Macroprep resin (Bio-Rad, Hercules, CA, USA) column [33] into four fractions SlF1-4. Fucoidan fractions F3 (SlF3) and F4 (SlF4) contained the highest amounts of fucose and galactose compared to fractions 1 (SlF1) and 2 (SlF2) (Appendix A). SlF3 and 4 were high molecular weight polysaccharides with average mass distribution ranging from 250 kDa to over 800 kDa (Appendix A). 

### 4.2. Chemical Analysis of Fucoidans 

Monosaccharide compositions of the fucoidans were analyzed as described previously [42]. Fucoidan fractions were hydrolyzed in 72% H_2_SO_4_ (5 mg/mL) at 30 °C for 1 h in water bath and then the mixture was diluted to 4% H_2_SO_4_ by adding water. The hydrolysis was continued for 40 min at 120 °C in autoclave. The hydrolysates were filtered through a 0.22 µm syringe filter and used for monosaccharide analysis [42]. Chromatographic separation was carried out at a flow rate 0.4 mL/min using the following three eluents: A-deionized water, B-200 mM NaOH and C-200 mM NaOH, 1 M NaOAc. The elution of neutral sugars was performed at 0.5% B in A for the first 17 min. Next, elution of uronic acid was performed by 3% B and 6% C in A for 20 min and completed with 100% B in 6 min. To calibrate the column, the 0.5% B in A was applied. The Dionex software Chromeleon™ 7.2 (Thermo Scientific, Waltham, MA, USA) was used for data quantification. Recovery values of the monosaccharides and uronic acid were estimated from runs at the same time.

The sulfate content was determined by the turbidimetric method [43]. In total, 110 μL hydrolysates after TFA hydrolysis were mixed with 120 μL 8% TCA. Then 60 μL 2% BaCl_2_ in 15% PEG6000 reagent was added. The mixture was allowed to stand for 35 min. The released BaSO_4_ suspension was measured at 500 nm in a microplate reader (TECAN Infinite 200, Salzburg, Austria). BaSO_4_ was used as standard to generate a linear standard curve for the sulfate response. 

The molecular weights (MW) of fucoidan fractions were determined by HP-SEC using an Ultimate iso-3100SD pump with WPS-3000 sampler (Dionex, Sunnyvale, CA, USA) connected to an ERC RefractoMax 520 refractive index detector (Thermo Scientific, Waltham, MA, USA) [23]. The fucoidan samples were prepared in 100 mM sodium acetate, pH 6 (3 mg/mL) and filtered through 0.22 µm filters. In total, 100 µL of samples were injected into a Shodex SB-806 HQ GPC column (300 × 8 mm) coupled with a Shodex SB-G guard column (50 mm × 6 mm) (Showa Denko K.K., Tokyo, Japan). Elution was carried out at a flow rate of 0.5 mL/min at 40 °C. External pullulan standards in the range of 305–805,000 Da (PSS Polymer Standards Service GmbH, Mainz, Germany) were applied to establish a polynomial relationship between the logarithmic molecular weight and the corresponding retention time in order to convert the retention times of the samples to molecular weights. Molecular weights above and below the applied standard range are estimates based on extrapolation of the polynomial model.

### 4.3. Sequence Analysis of the Mef2 Gene

The amino acid sequence of Mef2 (GenBank: ON099398; RefSeq: WP_055392200.1) was identified in the genome of *M. eckloniae* by BLAST using known GH107 fucoidanase encoding genes. The Clustal Omega service using HHalign algorithm (https://www.ebi.ac.uk/Tools/msa/clustalo/, accessed on 23 March 2022) [44] was used for multiple sequence alignments of the GH107 endo-fucoidanases. The α(1,4)-specific endo-fucoidanases MfFcnA (GenBank: CAI47003.1), FFA1 (RefSeq: WP_057784217.1), FFA2 (RefSeq: WP_057784219.1), Fp273 (GenBank: AYC81238.1), Fp277 (GenBank: AYC81239.1), Fp279 (GenBank: AYC81240.1) Fhf1 (RefSeq: WP_066217780) and Fhf2 (RefSeq: WP_066217784.1); the α(1,3)-specific endo-fucoidanses Fda1 (GenBank: AAO00508.1), Fda2 (GenBank: AAO00509.1) and L- fucoidanases P5AFcnA (GenBank: AYF59291.1), P19DFcnA (GenBank: AYF59292.1), SVI_0379 (GenBank: BAJ00350.1) were used for amino acid sequence comparisons. 

The signal peptide sequence and the protein domain predictions were performed using the SignalP5.0 server (http://www.cbs.dtu.dk/services/SignalP/, accessed on 17 June 2020) and the InterProScan V5 (https://www.ebi.ac.uk/interpro/, accessed on 29 March 2022), respectively. Pairwise alignments of Mef2 were performed using the MAFFT algorithm (EMBL_EBI, MAFFT < Multiple Sequence Alignment < EMBL-EBI) and Jalview software [45]. 

### 4.4. Construction and Cloning of the Expression Vectors

The construct containing the gene encoding Mef2 was designed to harbor a C-terminal 10×his tag. The synthetic gene, codon-optimized for *E. coli* expression, devoid of the predicted signal peptide and T9SS domain [28], was synthesized by GenScript (Piscataway, NJ, USA) and inserted into the pET-28b(+) vector between the NdeI and XhoI restriction sites.

### 4.5. Recombinant Enzyme Expression and Purification

The expression of the Mef2 fucoidanase was performed in *E. coli* BL21 (DE3) harboring the Pch2 (pGro7) plasmid (Takara Biolabs, Göteborg, Sweden). The LB broth contained 50 μg mL^−1^ kanamycin and 34 μg mL^−1^ chloramphenicol, while 0.05% (*w*/*v*) arabinose was used to induce expression of the pGro7 chaperone. The cells were cultured to reach OD_600_ 0.6–0.8 at 37 °C and 180 rpm before enzyme expression was induced by the addition of 1 mM isopropyl-β-D-1-thiogalactoside (IPTG). After 20 hours at 20 °C or for optimal expression at 4 h induction at 37 °C and 180 rpm, the cells were harvested by centrifugation at 8000× *g* for 15 min and 4 °C and were re-suspended in buffer (20 mM Tris-HCl, pH 7.4, 250 mM NaCl, 20 mM imidazole) at the ratio 1:3 (*m*/*v*). The enzyme was purified using a Ni^2+^ Sepharose HisTrap HP column resin (GE Healthcare, Uppsala, Sweden) as previously described [22]. A PD10 column (Sephadex G-25, GE Healthcare Uppsala, Sweden) was used to remove imidazole from the enzyme solution. Protein content was measured by the Bradford (Bio-Rad, Hercules, CA, USA) assay with bovine serum albumin as standard [46]. The enzymes were stored at −80 °C. Sodium dodecyl sulfate-polyacrylamide gel electrophoresis (SDS-PAGE; 12%) and western blotting, performed as previously described [22], were used to analyze the purity and size of the enzyme.

### 4.6. Endo-Fucoidanase Activity Assays

C-PAGE, using small quantities of fucoidan substrate (approximately 7 mg) and enzyme, was used to determine the optimal assay conditions for Mef2, while kinetic studies were conducted using FTIR analysis, using high amounts of fucoidans (approximately 700 mg, assay conditions are described in the FTIR section).

The Mef2 optimal reaction conditions were 0.12 mg/mL Mef2 in 20 mM Tris-HCl pH 8, 100 mM NaCl, 0.9% fucoidan, and 10 mM CaCl_2_. For full fucoidan hydrolysis, the reaction was performed at 35 °C for 24 h, while for investigating optimal reaction conditions the reactions were incubated for 2 h at 35 °C. The pH optimum determination was performed in reactions containing 20 mM buffer (UB4 buffer pH 2–8 or borate buffer pH 8–11). The temperature optimum was determined by running reactions at 20, 25, 30, 35, 37, 40, 45, 50, 55, 60, and 70 °C. Influence of divalent cations was investigated by first removing divalent cations by addition of 2 mM Ethylene DiamineteTetra acetic Acid (EDTA), desalting by PD10 followed by addition of 10 mM of different divalent cations (CaCl_2_, CuSO_4_, FeCl_2_, MgCl_2_, CoCl_2_, MnCl_2_, NiSO_4_, and ZnCl_2_). The influence of NaCl was investigated at different concentrations of 25, 50, 100, 150, 200, 250, 300, 350, 400, and 500 mM. Time-course experiments were performed from 0 min to 48 h. All reactions were stopped by heating at 80 °C for 10 min, and protein debris was pelleted by centrifugation (10,000× *g*, 15 min at 20 °C).

Before loading on the C-PAGE gel, the reaction was mixed with loading buffer (ratio reaction:buffer = 1:1) containing a 20% (*v*/*v*) solution of glycerol in water and 0.02% (*w*/*v*) phenol red. The samples (6–8 μL) were electrophoresed through a 20% (*w*/*v*) 1 mm thick resolving polyacrylamide gel in 100 mM Tris-borate buffer pH 8.3 at 30 mA for 90 min. Gel staining was performed with a solution containing 0.5% alcian blue 8 GX (Panreac, Barcelona, Spain) in 2% acetic acid and 0.02% O-toluidine (Sigma-Aldrich, Steinheim, Germany) in ethanol, for 1 h at room temperature. The gel was washed with distilled water until bands were visible.

### 4.7. Thermal Stability of Recombinant Fucoidanase Mef2

The thermal stability of the Mef2 protein was determined using dynamic light scattering (DLS). Mef2 (5 µM) was transferred to a 10 µL capillary tube and analyzed with a nanoDSF Prometheus NT.Plex instrument (Nanotemper-technologies, Munich, Germany). To obtain denaturation profiles, a temperature gradient of 25–80 °C with an increase of 1 °C min^−1^ was used to monitor thermal stability. The raw data were exported into datasets containing fluorescence between 330 and 350 nm (F330 and F350), as well as the ratio of these values (F330/F350) and absorbance at 350 nm (A350). Denaturation was visualized by plotting the first derivatives of F330/F350. The peak of the first derivative corresponds to the melting temperature (Tm), which is the transition midpoint of protein unfolding. Mef2 was treated with 2 mM EDTA followed by PD10 desalting to remove all EDTA. In total, 10 mM Ca^2+^ was afterward added for Tm assessments with Ca^2+^. The concentration of Mef2 (5 µM) was determined by measuring the absorbance at 280 nm using the calculated molar extinction coefficient computed by the ProtParam tool at ExPASy (https://web.expasy.org/protparam/, accessed on 10 December 2019).

### 4.8. Mef2 Kinetics by Fourier Transform InfraRed (FTIR) Spectroscopy Measurement and Parallel Factor (PARAFAC) Analysis

FTIR spectroscopy was used to monitor the degradation of fucoidans from *F. evanescens* and *S. latissima* [32]. Mef2 was dosed at (0.00), 0.38, 0.86, 1.71, 3.43, and 4.57 µM for reactions on fucoidans from *F. vesiculosus*, and at (0.00), 0.57, 1.14, 2.29, 3.43, and 4.57 µM for reactions on fucoidans from *S. latissima.*

All IR spectra were scanned using a MilkoScanTM FT2 FTIR instrument (Foss Analytical, Hillerød, Denmark) in the range from 1000–2000 cm^−1^. The cuvette was kept at 42 °C and had a path length of 50 m. A 1 mL reaction mixture containing 2% weight/volume (*w*/*v*) fucoidan in 0.02 M Tris-HCl buffer pH 7.4 and 100 mM NaCl and 20 mM CaCl_2_. Following the addition of the enzyme, each reaction mixture was injected directly into the cuvette, and 100 spectra were acquired in a row for the reactions with *F. vesiculosus* and *S. latissima* SlF4. To export the acquired spectral data, the Foss integrator (version 1.5.3, Foss Analytical, Hillerød, Denmark) was used. The data was then analyzed as described previously [32].

Parallel factor analysis (PARAFAC) was used to estimate calibration curves for all fucoidanase reactions. PARAFAC decomposition of the tensor X into three different matrices using one component. The spectral signal received from each enzyme reaction is represented by Matrix A (spectral mode loadings). The number of spectra received in a continuous period of time is represented by Matrix B, the distance between spectra is 16.6 s (time mode loadings), and Matrix C represents the relationship between the change in spectra in A and the different enzyme dosages in B. (the scores of PARAFAC). In this study, FTIR-PARAFAC was used to quantify endo-fucoidanase activity, as previously described [32].

### 4.9. Enzymatic Hydrolysis of Fucoidans and Product Separation 

The *S. latissima* fucoidan fraction 3 (SlF3) was deacetylated by dissolving 1.5 g of fucoidans in 150 mL 12% ammonia solution as previously described for fucoidan from *F. evanescens* [22]. The yield of deacetylated SlF3 (deSlF3) was 89.3%. The Mef2 enzymatic hydrolysis of deSlF3 was performed with 10 g/L substrate, 10 mM CaCl_2_, 0.5 mg/mL Mef2 enzyme in 20 mM Tris-HCl buffer pH 8, and 100 mM NaCl at 35 °C for 24 h. The reaction was stopped by heating at 80 °C for 10 min and the precipitated enzyme was removed by centrifugation at 19,000× *g* for 15 min. The products were separated into two fractions, the medium molecular weight products (MMP) and the low molecular weight products (LMP) by adding cold ethanol 96% with a ratio of 3:1 (*v*/*v*) and collected by centrifugation at 15,000× *g* for 45 min. The ethanol in the supernatant was evaporated, thus isolating the low molecular weight products (LMP). 

Further separation of the LMP was performed by applying 0.6 g/L LMP in water to a Q sepharose high-performance resin column (1 cm × 20 cm) equilibrated with water. The oligosaccharides were eluted at a flow rate 0.7 mL/min by a linear gradient of ammonium bicarbonate salt in water from 0 to 2 M. The total carbohydrate content in fractions was analyzed by the phenol-sulfuric acid method [47]. In total, 20 µL aqueous phenol solution 5% (*w*/*v*) was added to 20 µL fucoidan fraction followed by addition of 200 µL concentrated sulfuric acid. The absorbance was measured at 490 nm. Fractions containing carbohydrates were further analyzed by C-PAGE. The fractions containing the same oligosaccharides on C-PAGE were combined and ammonium bicarbonate salts were removed by evaporation and the samples were lyophilized. 

Further separation of the MMP was performed by DEAE-Macroprep resin (Bio-Rad, CA, USA) column using a linear gradient of NaCl from 0.1 to 2 M. The fractions were combined based on total carbohydrate content [47] and C-PAGE analysis. The MMP-derived fractions were dialyzed over water using a 3.5 kDa dialysis tube (Thermo Fisher Scientific, Waltham, MA, USA), to remove salts, and lyophilize. 

### 4.10. NMR Spectroscopy

Native *S. latissima* fucoidans, as well as LMP, MMP, and all the further separated oligosaccharides were analyzed by NMR spectroscopy. The samples (approximately 10 mg) were dissolved in 500 μL ^2^H_2_O, and NMR spectra were collected on an 800 MHz Bruker Avance III HD instrument equipped with a 5 mm TCI cryoprobe and a SampleJet sample changer. Oligosaccharide spectra were acquired at 25 °C. The ^1^H 1D NMR spectra (of 16,384 complex data point sampling 1.7 s) were acquired by summing up 16 transients. The ^1^H-^1^H TOCSY (2048 × 400 complex data points sampling 128 ms and 25 ms in the direct and indirect dimensions, respectively) was acquired by using a 10 kHz spin lock field during a mixing time of 80 ms. The ^1^H-^1^H COSY was acquired by sampling 2048 × 512 complex data points for 213 ms and 53 ms in the direct and indirect dimensions, respectively, while ^1^H-^1^H NOESY was acquired as 2048 × 256 complex data points by sampling the FID for 213 ms and 53 ms in the direct and indirect dimensions, respectively, and using a mixing time of 600 ms. The ^1^H-^13^C HMBC (2048 × 128 complex data points sampling 256 ms and 6.3 ms, respectively) and multiplicity-edited ^1^H-^13^C HSQC using adiabatic decoupling (2048 × 512 complex data points sampling 213 ms and 15.5 ms) were acquired alongside ^1^H-^13^C HMBC (1024 × 100 complex data points sampling 128 ms and 3 ms). Assignment spectra of the high molecular weight fractions were acquired at 50 °C with an 800 MHz Bruker Avance III instrument equipped with a 5 mm TCI cryoprobe and an Oxford magnet. All NMR spectra were processed with ample zero filling in all dimensions and baseline correction using Bruker Topspin 3.5 pl7 software. All spectra were subsequently analyzed with the same Bruker Topspin software.

## 5. Conclusions

In the present study, the novel endo-fucoidanase Mef2, from the marine bacterium *M. eckloniae*, was characterized. Mef2 is the first enzyme of the GH107 family to show activity on the highly complex fucoidans of *S. latissima*. In *S. latissima* fucoidans, Mef2 specifically catalyzed hydrolysis of sulfated α(1,3)-linked fucosyl residues with α(1,4)-linked fucosyl branches. These fucoidan oligosaccharide structures with α(1,4) branches have not previously been described for fucoidans from *S. latissima,* and they are the first fucoidan oligosaccharides from *S. latissima* released by a fucoidanase. 

## Figures and Tables

**Figure 1 marinedrugs-20-00305-f001:**
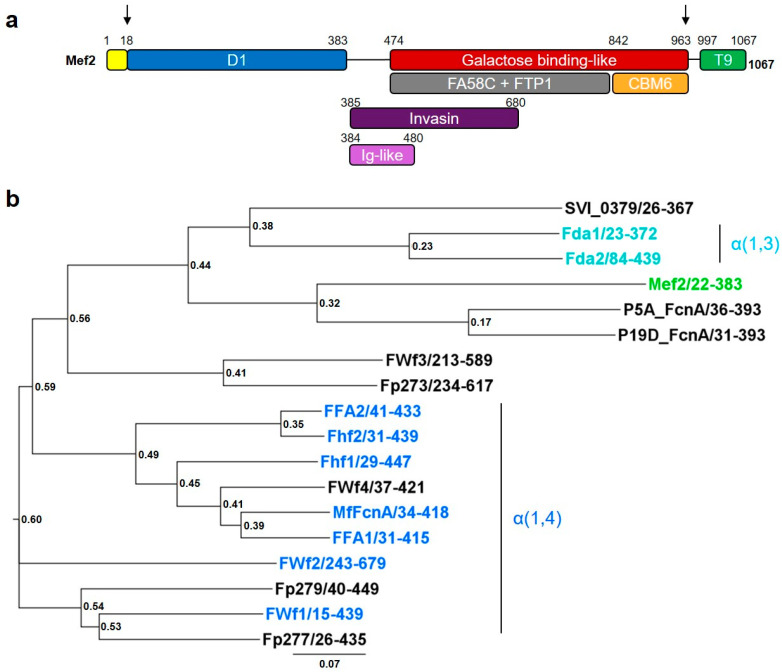
Modular structure of the native Mef2 and recombinant Mef2 protein and phylogenetic overview. (**a**) The native Mef2 protein sequence, arrows indicate the recombinantly expressed part of the Mef2 protein. Yellow: signal peptide, blue: D1 catalytic domain, purple: invasin/intimin cell-adhesion domains (IPR008964), pink: Ig-like domain (IPR003343), red: galactose binding-like domain (IPR000421), grey: FA58C domain (IPR000421) and FTP1 domain (IPR006585), orange: CBM6 domain (IPR005084), and green: secretion system C-terminal sorting domain (T9SS domain (T9)) (IPR026444). Domains were predicted using SignalP (signal peptide), InterProScan, and sequence alignment (D1) with P5AFcnA and MfFcnA. (**b**) phylogenetic analysis of the D1 domain of selected GH107 fucoidanases (numbers indicate the used sequence span), blue: α(1,4)-linkage specific fucoidanases, Turquoise: α(1,3)-linkage specific fucoidanases, green: Mef2. Accession numbers can be found in Appendix A.

**Figure 2 marinedrugs-20-00305-f002:**
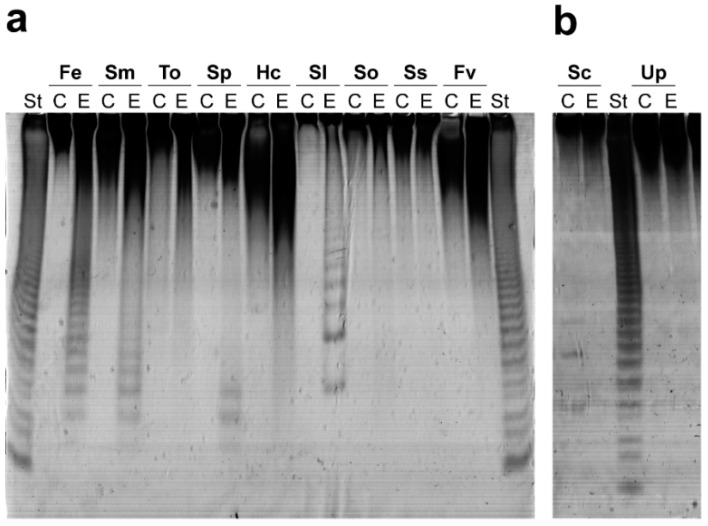
Substrate specificity of Mef2 on the fucoidans from different brown seaweeds. C-PAGE of (C) substrate control and (E) Mef2 reaction on different fucoidans from brown seaweeds (**a**) *F. evanescens* (Fe), *S. mcclurei* (Sm), *Turbinaria ornata* (To), *S. polycystum* (Sp), *Hormophysa cuneiformis* (Hc), *S. latissima* (Sl), *Sargassum oligocystum* (So), *Sargassum serratum* (Ss), *Fucus vesiculosus* (Fv), (**b**) *S. cichorioides* (Sc), and *Undaria pinnatifida* (Up). St) reaction of FFA2 on fucoidans from *F. evanescens*.

**Figure 3 marinedrugs-20-00305-f003:**
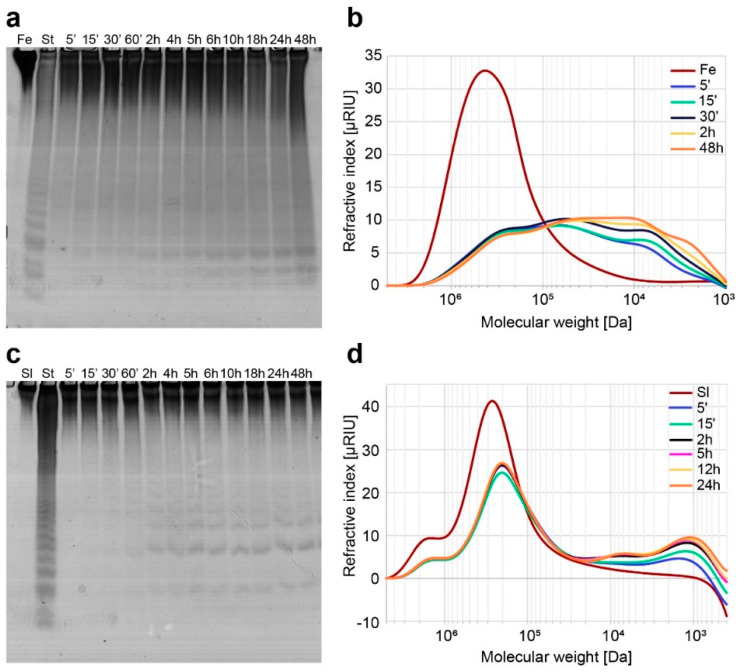
Mef2 catalyzed hydrolysis of fucoidans by C-PAGE and HP-SEC analysis. C-PAGE analysis of Mef2 on fucoidans from (**a**) *F. evanescens* (Fe) and (**c**) *S. latissima* (Sl), from 5 min to 48 h of reaction. HP-SEC chromatograms of Mef2 hydrolysis on the fucoidans from (**b**) *F. evanescens* and (**d**) *S. latissima*, from 5 min to 48 h. (St) *F. evanescens* fucoidans hydrolyzed by FFA2.

**Figure 4 marinedrugs-20-00305-f004:**
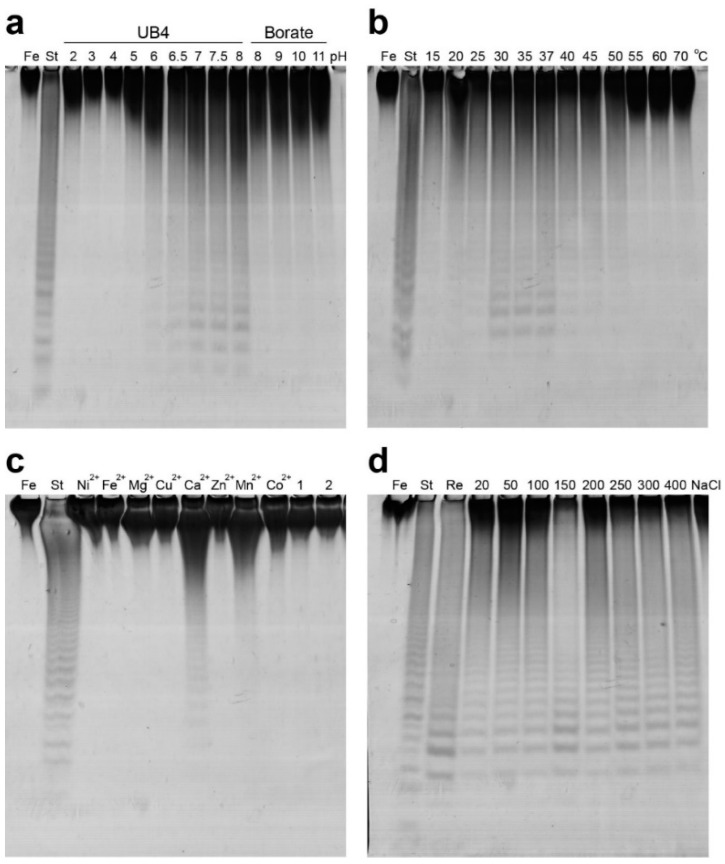
Mef2 optimal conditions on fucoidans from *F. evanescens*. C-PAGE analysis of Mef2 activity at different pH (**a**), temperature (**b**), divalent cations (**c**), and NaCl concentrations (**d**). (St) Reaction of FFA2 on fucoidans from *F. evanescens*. (Fe) *F. evanescens* fucoidan substrate.

**Figure 5 marinedrugs-20-00305-f005:**
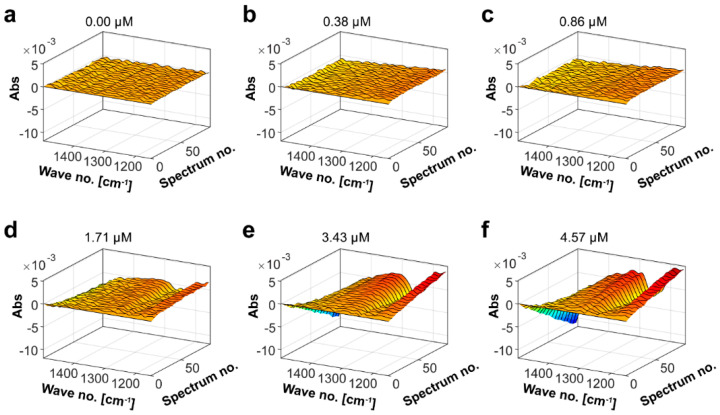
Mef2 kinetics on fucoidans from *F. evanescens* using FTIR. Spectral evolution profiles for the Mef2 endo-fucoidanase using 2% *w*/*v* of *F. evanescens* fucoidans (FeF4) using different enzyme dosages: (**a**) 0.00 µM, (**b**) 0.38 µM, (**c**) 0.86 M, (**d**) 1.71 µM, (**e**) 3.43 µM and (**f**) 4.57 µM. The spectral evolution depends on the enzyme concentration. The spectral changes of buffer alone and substrate alone were subtracted. Time per spectrum was 16.6 s [32]. Chemical composition of FeF4 is shown in Appendix A.

**Figure 6 marinedrugs-20-00305-f006:**
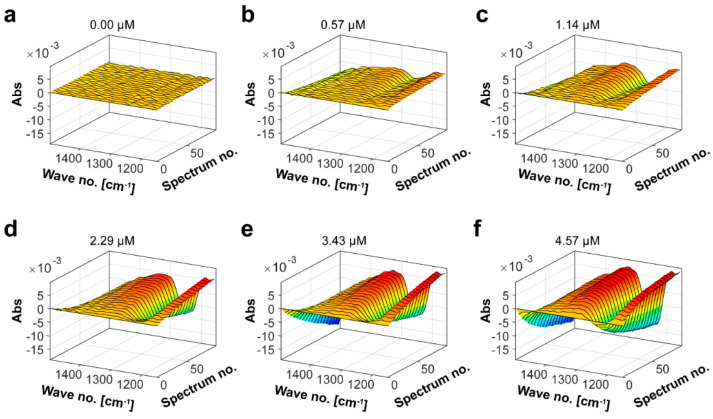
FTIR spectral evolution profiles for Mef2 activity on fucoidans from *S. latissima*. SlF4 hydrolysis using different Mef2 enzyme dosages: (**a**) 0.00 µM, (**b**) 0.57 µM, (**c**) 1.14 µM, (**d**) 2.29 µM, (**e**) 3.43 µM and (**f**) 4.57 µM. The spectral evolution depends on the enzyme concentration. The spectral changes of buffer and substrate were subtracted. Time per spectrum was 16.6 s [32]. Chemical composition of SlF4 is shown in Appendix A.

**Figure 7 marinedrugs-20-00305-f007:**
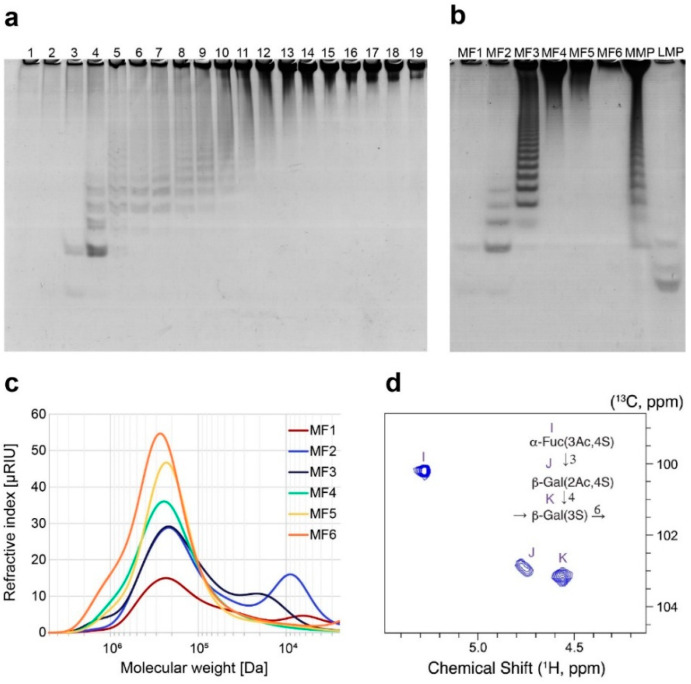
Analysis of Mef2 released *S. latissima* medium molecular weight products (MMP) and further separated fractions (MF). Medium molecular weight products (MMP) after Mef2 hydrolysis was further separated into (**a**) 19 fractions (1–19), which were pooled when the same oligosaccharides were present in C-PAGE. (**b**) C-PAGE of pooled and purified medium molecular weight fractions MF1-MF6 compared to LMP and MMP. (**c**) HP-SEC chromatogram of MF1-6. Pullulan was used as standard. (**d**) Anomeric region of the ^1^H-^13^C HSQC spectrum for the acetylated MMP. The basic structure of the MMP as determined by NMR yields the substitutions shown schematically and is consistent with the high galactose content in this fraction and with previous structural analyses of *S. latissima* fucoidans [3]. The ^1^H-^1^H TOCSY NMR spectrum for the deacetylated MMP is shown in Appendix A.

**Figure 8 marinedrugs-20-00305-f008:**
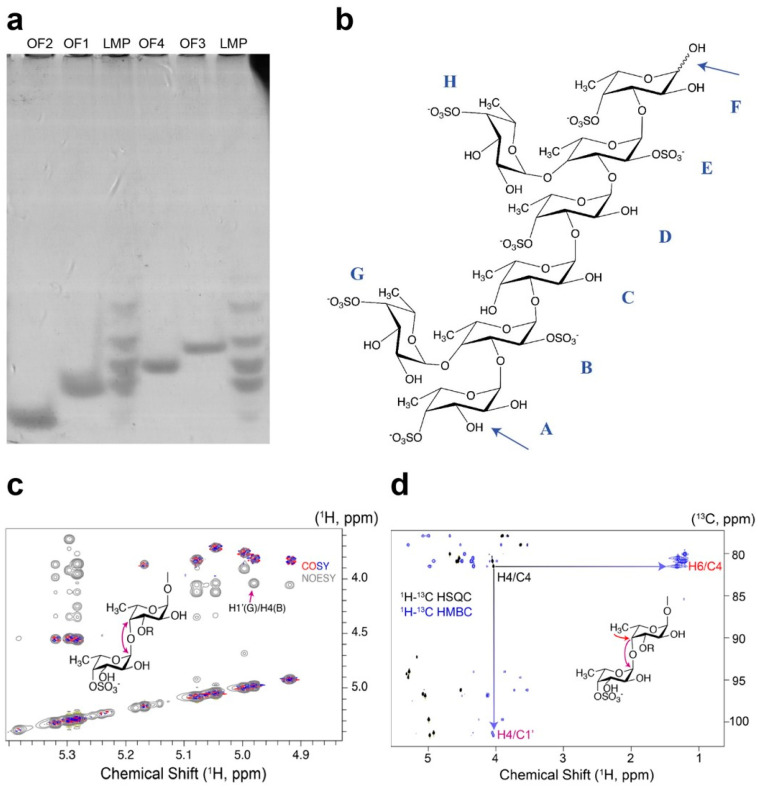
Analysis of Mef2 released LMP from *S. latissima* fucoidans. (**a**) C-PAGE of purified oligosaccharide fractions OF1 to OF4. (**b**) Molecular structure of the purified octa-saccharide OF1. Units have the same identifier as in Table 4; blue arrows indicate Mef2 cleavage sites. Structure determination of the α(1,3)-linked fucoidan backbone with the 1-4 linked sulfated fucosyl substitution of the OF1 oligosaccharide. (**c**) Overlay of ^1^H-^1^H COSY and ^1^H-^1^H NOESY spectra, showing the NOE across the glycosidic bond for hydrogens 1 and 4 of residues G and B. (**d**) Overlay of ^1^H-^13^C HSQC and ^1^H-^13^C HMBC spectrum showing the correlation across the glycosidic bond for the well-resolved signals of the 1-4 linked residues G and B.

**Table 1 marinedrugs-20-00305-t001:** Alignment of the catalytic amino acids in Mef2 and other GH107 members. Red: the active site aspartate (D), blue: the active site histidine (H), and orange: the change in the conserved arginine to valine in Mef2. Protein accession numbers are listed in Appendix A.

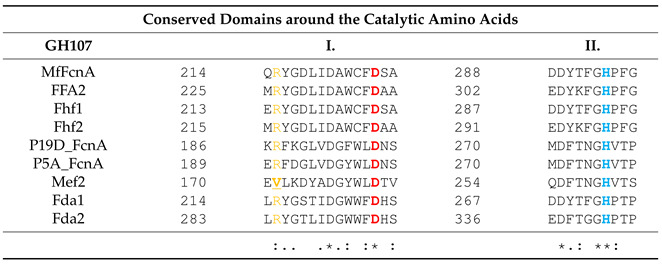

Below the alignment, an * indicates positions in the alignment which have a fully conserved residue. A colon (“:”) indicates conservation between amino acids of strongly similar properties, and a period (“.”) indicates conservation between groups of amino acids of weakly similar properties. I and II indicates the investigated protein domains.

**Table 2 marinedrugs-20-00305-t002:** Alignment of the −1 subsite in Mef2 and other GH107 members. Purple: conserved amino acids in the −1 subsite and orange: the change in the conserved asparagine to serine in Mef2. Protein accession numbers are listed in Appendix A.

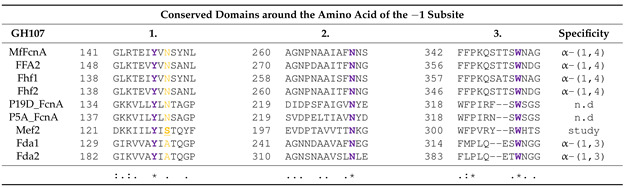

Below the alignment, an * indicates positions in the alignment which have a fully conserved residue. A colon (“:”) indicates conservation between amino acids of strongly similar properties, and a period (“.”) indicates conservation between groups of amino acids of weakly similar properties. 1, 2 and 3 indicates the investigated protein domains.

**Table 3 marinedrugs-20-00305-t003:** ^1^H and ^13^C NMR data for the deacetylated medium molecular weight products (MMP) purified after cleavage of *S. latissima* fucoidans with endo-fucoidanase Mef2. Acetylations as determined on native samples are indicated in italic font.

Residue		Chemical Shifts (ppm)					
		H1/C1	H2/C2	H3/C3	H4/C4	H5/C5	H6/C6
I	α-L-Fuc*p*(*3Ac,* 4SO_3_^−^)-(1→	5.282	3.785	4.099	4.589	4.510	1.258
		100.3	69.0	68.6	81.1	66.5	15.8
J	-3)-β-D-Gal*p*(*2Ac,* 4SO_3_^−^)-(1→	4.773	3.731	3.908	4.692	3.798	3.802
		102.8	71.6	76.5	77.3	74.4	61.0
K	-4,6)-β-D-Gal*p*(3SO_3_^−^)-(1→	4.580	3.815	4.426	4.540	3.930	4.188/3.924
		103.1	68.9	79.9	73.8	73.2	70.3

**Table 4 marinedrugs-20-00305-t004:** ^1^H and ^13^C NMR data for the purified low molecular weight product fraction OF1 after Mef2 cleavage of *S. latissima* fucoidans. Both the reducing end F and the adjacent residue E exhibit two sets of signals due to the anomeric forms of the reducing end.

Residue\Atom		Chemical Shifts (ppm)					
		H1/C1	H2/C2	H3/C3	H4/C4	H5/C5	H6/C6
A	*α*-L-Fuc*p*(4SO_3_^−^)-(1-	5.036	3.71	3.982	4.561	4.458	1.211
		96.87	68.5	69.06	80.87	66.6	15.84
B	→3,4)-*α*-L-Fuc*p*(2SO_3_^−^)-(1-	5.283	4.556	4.122	4.037	4.331	1.325
		94.05	73.2	73.2	81.46	67.96	15.47
C	→3)-*α*-D-Fuc*p-*(1-	4.995	3.764	3.858	3.954	4.25	1.181
		99.81	66.96	75.67	68.62	66.7	15.59
D	→3)-*α-* D-Fuc*p*(4SO_3_^−^)-(1-	5.076	3.833	3.905	4.68	4.475	1.211
		96.74	67.53	77.9	80.08	66.54	15.84
Eα	→3,4)-*α-* D-Fuc*p*(2SO_3_^−^)-(1-	5.32	4.552	4.12	4.05	4.34	1.29
		94.28	73.2	73.2	80.85	67.95	15.65
Eβ	→3,4)-β*-* D-Fuc*p*(2SO_3_^−^)-(1-	5.297	4.545	4.114	4.054	4.35	1.29
		94.69	73.2	72.8	80.85	67.95	15.65
Fα	→3)-α*-* D-Fuc*p*	5.161	3.872	3.871	3.986	4.118	1.165
		92.25	66.56	75.36	68.41	66.07	15.61
Fβ	→3)-β*-* D-Fuc*p*	4.517	3.539	3.633	3.92	3.72	1.203
		96.25	70.12	78.93	68.1	70.67	15.583
G	α*-* D-Fuc*p*(4SO_3_^−^)-(1→	4.974	3.824	4.103	4.584	4.37	1.254
		101.6	69.05	68.03	80.95	67.172	16.14
H	α*-* D-Fuc*p*(4SO_3_^−^)-(1→	4.918	3.831	4.074	4.546	4.338	1.307
		101.34	69.38	68.37	80.55	67.33	16.38

## Data Availability

Not applicable.

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
