# Peer review of "The Endo-α(1,3)-Fucoidanase Mef2 Releases Uniquely Branched Oligosaccharides from Saccharina latissima Fucoidans"

_marinedrugs, 2022, doi:10.3390/md20050305_

Round 1

Reviewer 1 Report

  1. In this study, a novel Endo-α(1,3)-fucoidanase Mef2 was 

    characterized.  Mef2 could degrade highly complex fucoidans from S. latissima and release branched oligosaccharides. This study is interesting. Because of the wide variety of fucoidan structures in nature, fucoidanases specific for different structures including differeent glycosidic linkage, sulfation position and branching patterns are necessary for the structural elucidation and preparation of particular fucoidan oligosaccharides.  

  2. From Table S5, sulfate contents of MF1-MF4 range from 48.2% to 54.8%. These data are very high. But the high degree of sulfation was not reflected in NMR spectra. So it is necessary to verify the exact sulfation pattern by other analytical methods.
  3. The neutral monosaccharide composition of MF1-MF6 was shown in Table S5. Are there any uronic acids present in these fractions?
  4. The structure of OFs were mainly elucidated through NMR spectra. What about their monosaccharide composition and sulfate content?

Author Response

Dear Reviewer 1, 

Thank you for your questions and suggestions, here are our replies:

1. no reply needed

2. We observed an almost 1:1 ratio of sulfation:monosaccharide (1 sulfate per sugar residue) in the MF NMR (I, J and K) and since the sulfate content is given in mol% of the monosaccharides, it corresponds well with a mass fraction near 50%. We note this approximate sulfation degree has been found previously for native fucoidan, extracted by the aid of enzymes (Nguyen et al. 2020. Marine Drugs 18)

3. Yes, there are small amounts of uronic acids in the MF1-6 fractions. We have added the determination of the uronic acids GuluA, GluA and ManA in Table S6 (Tables and figures in the supplementary material has changed in the revised manuscript).

4. Fucose was the only monosaccharide detectable in the LMP fraction, consistent with the findings through NMR analysis. We have added a Table (Table S5) of the composition of the MMP fraction and we have added a line in the text explaining this (line 350-352).

Reviewer 2 Report

In this manuscript, a new endo-fucoidanase Mef2, from the marine bacterium Muricauda eckloniae, was characterized by an array of detailed experiments. Interestingly, Mef2 specifically catalyzed hydrolysis of α(1,3)-linked fucosyl residues with α(1,4)-linked fucosyl branches identified in fucoidan from Saccharina latissima, which is the first enzyme of the GH107 family to show such activity. This manuscript worth reading, however, improvements are required.

  1. Page 3 Lines 113–114: Pease use references’ numbers instead of ‘Colin et al., 2006a; Silchenko et al., 2017a, 2017b; Vuillemin et al., 2020; Trang et al., 2022’.
  2. There is no green character in Tables 1 & 2, only observed in Figure S2.
  3. It is interesting that expression of Mef2 at 20 °C overnight resulted in a partially degraded protein, however the degradation didn’t happened when expressed for 4 hours at 37 °C. Could you give a explanation, please?
  4. Page 5 Line 186: Pease use a reference number instead of ‘Ale et al., 2011’. This phenomenon happened again on Page 9 Line 306 (‘Vy et al., 2022’). Perhaps this is a technical issue. Please check this issue throughout the whole manuscript.
  5. In Figure 2 caption, there is mistake for the notes. There are characters C and E, not numbers 1 and 2. In Figure 2 caption, there is mistake for the notes. There are characters C and E, not numbers 1 and 2. It is mentioned ‘Mef2 also catalyzed hydrolysis of the galactofucans isolated from S. mcclurei, Saccharina cichorioides and S. latissima’ on Page 5 Line 189. However, compared the bands of E sets of Sm (S. mcclurei) and Sl (S. latissima) in Figure 2a, there is no obvious low molecular weight products bands of Sc (S. cichorioides) in Figure 2b. After rechecking the Figure 2a, two hydrolysis products bands could be observed for Sp (S. polycystum).
  6. On Page 5 Line 189, ‘the complete degradation was not achieved before 5 hours on fucoidans from S. latissima’. To keep consistency, the HP-SEC chromatogram of Mef2 hydrolysis on the fucoidans from S. latissima for 5 hours should be added in Figure 2d.
  7. Please define the ‘Tm’ as the abbreviation of ‘Temperature’ when it appeared for the first time on Page 7 Line 240. And please revise ‘Ca2+’ as ‘Ca2+’ In Figure S4 caption.
  8. What is the composition of F. evanescens fucoidan (FeF4) used in the Mef2 kinetics experiments?
  9. The slops for the same equation were different on Page 9 Lines 296 and 299.
  10. On Page 9 Lines 303–306, There is a mistake for the conclusion after the comparison of the specific activity values. The Mef2 enzyme presumably works faster than Fhf2, not slower.
  11. Please add Table S2 after SIF4 in Figure 6 caption and on Page 10 Lines 317, to tell the readers the composition of SIF4.
  12. In Figure S8b, the average molecular weight of MMP is larger than that of SI. Why?
  13. In Figure 7b&C, the C-PAGE analysis was contradictory to the HP-SEC chromatogram. For example, the average molecular weight of MF1 was smaller than that of MF5 as indicated in the C-PAGE, but it was a bit larger than that of MF5 in the HP-SEC chromatogram.
  14. As recorded in Figure 7 caption, Figure 7d is a partial of the 1H-13C HSQC spectrum, the coupling pattern between H1 and H4 can not be deduced from this spectrum. The 1H-1H COSY or 1H-1H TOCSY spectrum should be added here, or in the Supplementary Material.
  15. The NMR data on Page 12 Lines 384–388 were not consistent with those shown in Figure 7d.
  16. How to judge that OF1 was the purest among the four fractions OF1–4? The C-PAGE in Figure 8a showed the clear one band for OF3 and OF4 respectively, and both the bands were much narrower than the one of OF1.
  17. There are two sets of NMR data for the residues E and F in Table 4. Do anomeric isomers coexist?
  18. Page 13 Lines 431&434: 1H-13C HMBC spectra →1H-13C HMBC spectrum

Author Response

Dear Reviewer 2, 

Thank you for all your valuable questions and suggestions. Here are our replies:

1. Thank you for noticing this issue, we have corrected it through the text

2. We agree that the green color was very hard to see in the Tables and also in the alignment in Figure S2 (S1 in the revised manuscript). We have changed the color to orange and colored the whole column in both tables and in the alignment. We have furthermore underlined the Mef2 amino acids in question.

3. Yes, this is interesting and can occasionally be observed for enzymes. In the case of Mef2 it might be related to an in E. coli time-dependent unfolding of the Mef2 enzyme, resulting in signaling for protein degradation, although this is merely a hypothesis and we have not investigated this further. We have added a sentence in the text in line 184-186.

4. Thank you for noticing this issue, we have corrected it through the text

5.:

5.a 1 and 2 have been changed to C and E, respectively

5.b We have improved the image quality, so the oligosaccharide bands of Mef2 hydrolysis of fucoidan from S. cichorioides can be clearly visualized.

5.c We thank the reviewer for noticing the hydrolysis of fucoidan from S. polycystum, this has now been added to the text in line193

6: We have added a new Figure 3, where 5 hours have been added to Figure 3d (we think the reviewer means Figure 3, since there are no HP-SEC in Figure 2)

7:

7.a Tm has been written out the first time, thank you

7.b Ca2+ has been changed to Ca2+

8: Thank you for making us aware that evidence has not been provided for the composition of this fraction. We have included a supplementary Table (Table S3 in the revised manuscript) with the composition as well as written a sentence in the methods line 601 and references in the text in line 298 as well as in the Figure 5 Figure caption.

9: We thank the reviewer for notifying us of this mistake, it has now been corrected

10: We thank the reviewer for this observation, yes, indeed the Mef2 enzyme is faster not slower, we have corrected this in the text.

11: Thank you for the advice, we have added Table S4 (which is the old Table S2; the tables and figures have been shifted numbers in the supplementary in the revised manuscript) references as suggested. We have also included a reference to the FeF4 composition Table S3 in the same way for consistency.

12: We very much appreciate this observation, the sample names of Sl and MMP had been switched, we have now corrected it.

13: The C-PAGE and HP-SEC results cooporate each other. We have rephrased the text to clarify our observations and interpretations from line 363. The high molecular weight fucoidan (around 200 kDa) cannot migrate in the C-PAGE gel, but will be retained in the top of the C-PAGE gel. For MF4 and 5 the high molecular weight fucoidans however migrate slightly into the gel, resulting in a smear in the top of the gel, suggesting slightly lower molecular weight and this is likely what is observed in the shift to the right (lower molecular weight) of the major peak in the HP-SEC for these two samples. For MF1 the high molecular weight fucoidans are not migrating in the C-PAGE gel, but the low molecular weight fucoidans are, resulting in visible bands in the gel and this is supported by a peak in the HP-SEC around 4-10 kDa.

14: Thank you for bringing this matter to our attention. Due to the complexity of the full spectrum, Figure 7 focuses on the anomeric signals (i.e. the acetal C1H1 groups that appear in a spectral region that is remote from all alcohol CH signals) of the major units as shown in the inset. We have now added a new Figure with the TOCSY spectrum (Figure S9) that is consistent with the data from Table 3 on the deacetylated preparation, as suggested. The TOCSY shows the correlates between H1 and hydrogens 2, 3 and 4 in each unit. The main text and Figure 7 caption now refers to the new supplementary Figure S9 upon mention of the spin systems.

15: Figure 7d only shows the the H1/C1 coordinates, and the data is consistent with the values for H1/C1 in Table 3; the chemical shift values in lines 384-388 are however not consistent with Table 3, as in lines 384-388 refer to acetylation sites in the native sample, while Table 3 reports the chemical shifts for the more homogeneous deacetylated fraction. We have tried to clarify that Table 3 reports data on the deacetylated preparation.

16: We agree that this was unclearly written, all samples were of high purity, we have adjusted the text.

17: Yes, that is fully correct, we specify in Table 4 “Both the reducing end F and the adjacent residue E exhibit two sets of signals due to the anomeric forms of the reducing end.”

18: We have corrected the wrong plural thrice to now state:

“Figure 8 exemplifies the identification of the α(1,4) linked branch from 1H-13C HSQC and 1H-13C HMBC spectrum. Beyond correlating to C5 atoms with characteristic chemical shifts near 67 ppm, the methyl protons at C6 also correlate to C4 in the 1H-13C HMBC spectrum. Two characteristic C4 positions can be identified with deshielded 13C nuclei (80.85-81.46 ppm), but non-deshielded 1H, indicative of glycosidically linked C4 positions. These C4 positions accordingly show 3JCH correlations in the 1H-13C HMBC spectrum to anomeric CH groups of other spin systems (highlighted for one of the residues with branching at C4 in Figure 8c,d

Round 2

Reviewer 1 Report

In this study, a novel Endo-α(1,3)-fucoidanase Mef2 was characterized. Mef2 could degrade highly complex fucoidans from S. latissima and release branched oligosaccharides. This study is interesting. 

The authors have revised the manuscript according to the reviewers's comments. The present version can be accepted for publication.

Reviewer 2 Report

The comments has been addressed and the paper is revised accordingly. No further suggestion from me.